# Effects of Inorganic Passivators on Gas Production and Heavy Metal Passivation Performance during Anaerobic Digestion of Pig Manure and Corn Straw

**DOI:** 10.3390/ijerph192114094

**Published:** 2022-10-28

**Authors:** Xiaoliang Luo, Bincheng Zhao, Mingguo Peng, Rongyan Shen, Linqiang Mao, Wenyi Zhang

**Affiliations:** School of Environmental Science and Engineering, Changzhou University, Changzhou 213164, China

**Keywords:** pig manure, anaerobic digestion, gas production, inorganic passivator, heavy metals

## Abstract

The treatment of livestock manure caused by the expansion of the breeding industry in China has attracted wide attention. Heavy metals in pig manure can pollute soil and water and even transfer to crops, posing harm to humans through the food chain. In this study, corn straw was selected as the additive and introduced into the anaerobic digestion. Sepiolite (SE), ferric oxide (Fe_2_O_3_), attapulgite (AT) and ferric sulfate (FeSO_4_) were used as passivators to compare the effects of these inorganic passivators on gas production and passivation of heavy metals during the process of the anaerobic digestion. When the dry mass ratio of pig manure to straw is 8:2, the gas production efficiency is optimal. SE, AT and ferric sulfate have a much stronger ability to improve gas production performance than Fe_2_O_3_. The total gas production increased by 10.34%, 6.62% and 4.56%, and the average methane production concentration increased by 0.7%, 0.3% and 0.4%, respectively. The influence of SE, AT and ferric sulfate on the passivation of heavy metals is much better than Fe_2_O_3_, and the fractions in biological effective forms of Cu and Zn reduced by 41.87 and 19.32%, respectively. The anaerobic digestion of mixed materials is conducive to the gas production and the passivation of heavy metals. Therefore, SE, AT and ferric sulfate are selected as composite passivators, and the optimal ratio of inorganic composite passivators i: AT 7.5 g/L, ferric sulfate 5 g/L and SE 7.5 g/L, according to the results of orthogonal experiments. This study can provide a theoretical basis for the safe application of biogas fertilizers.

## 1. Introduction

With the development of social economy and the improvement of people’s living standards, the annual output of livestock and poultry manure in China has reached 3.8 billion tons in recent years [1]. The number of Chinese pig stock reached 442 million by 2016 [2]. The large scale of the pig breeding industry has brought a series of environmental problems in terms of water pollution and air pollution [3], especially the environmental pollution caused by pig manure [4]. Untreated pig manure can produce large amounts of noxious gases, and pig manure can easily penetrate into the ground due to rain erosion, polluting soil and groundwater and entering water bodies with underground runoff, thus polluting water resources [5]. In order to improve the efficiency of farming, a large number of heavy metals are added to the animal feed in the modern husbandry industry, which exceeds the animal’s absorption capacity and is excreted in the form of feces and urine. In 2017, the total heavy metals sourced from manures was 2.86 × 10^5^ t with the predominant contribution originating from pig manure (71.52%) [6] posing a very high pollution risk. As, Cr, Hg and Cd in livestock feces are of low content but of great toxicity [7]. Heavy metals gather in animals and plants through the food chain. The high concentration of Cu and Zn in the untreated manure will be accumulated in plants by their entrance to the environment. The efficiency of photosynthesis decreases with the excessive accumulation of Cu and Zn and plant growth is inhibited [8]. The original edible and medicinal value of plants is also reduced; for example, the antinociceptive activity of guava is limited by the excessive accumulation of Zn and Cu [9]. Excessive intake of Cu and Zn will damage the stomach, cardiovascular and nervous system [10]. In addition, excessive intake of Zn inhibits the absorption of Cu and Fe and excessive Cu damages protein, lipids and DNA [11]. The harm to the environment caused by pig manure is an environmental problem that needs to be solved urgently, and the rational and full utilization of pig manure resources is an emerging industry with great development prospects.

The anaerobic composting of pig manure can not only effectively solve the problem of pig manure accumulation but also obtain biogas and other substances, which is a common treatment method. However, traditional composting and aluminum sulfate treatment brings many hidden dangers of waste and secondary pollution [12]. Heavy metals can inhibit the activity of microorganisms during anaerobic digestion and reduce gas production [13]. Many anaerobic composting additives also have a certain negative effect on methane production. Zhao et al. found that the addition of biochar and manganese sulfate to the anaerobic composting of pig manure could inhibit the activity of methanogens and lead to the decline of methane production [14]. The study of Ren et al. also proved that the addition of clay had a similar effect [15]. Hu et al. found that adding a filamentous microalgae to the dry anaerobic digestion of pig manure could improve the methane production of the fermentation system [16], but microalgae additives had the problems of high cost and low energy conversion efficiency [17]. The advantages of the dry co-digestion of food waste and pig manure are the low cost of kitchen waste and high energy conversion rate, but there is also the problem of high fatty acids inhibiting methane production [18].

As an agriculturally developed country, many crops, such as rice, bamboo, wheat and corn, are planted in China, which produces much surplus straw. Due to the relatively high cost of traditional straw treatment and the large amount of harmful gases produced by open-air incineration (NO_2_, CO, SO_2_) [6], a new technology needs to be developed. Anaerobic digestion is a simple, reliable and environmentally friendly technique which is not only treats solid waste reasonably well but also increase methane production to truly realize a circular economy by combining livestock manure, crop straw and inorganic minerals. Straw itself can produce biogas under anaerobic digestion [19]. Many scholars have confirmed that anaerobic digestion can effectively promote the gas production of the system by controlling the dry matter of the crop straw and livestock manure to a certain level. For example, Zhao et al. conducted the anaerobic digestion of pig manure and oat straw in different proportions. They found that mixed anaerobic digestion could improve the gas production when the C/N ratio was 27 [20]. Xiang et al. used rice straw and pig manure as raw materials for anaerobic digestion and found that the methane production of the test group was improved with the addition of rice straws which were treated by alkaline and ultra-sonication [21]. Li et al. found that corn stalks pretreated with NaOH had higher methanogenic performance under the condition of high organic loading rate and 40 °C [22].

Inorganic passivators are also used to improve the gas production performance and passivate the heavy metals of the anaerobic digestion of livestock manure. Clay minerals, such as attapulgite (AT) and sepiolite (SE), have high specific surface areas, developed pore structure and unique crystal structure [23]. Van der Waals force (physical adsorption), chemical bond force (chemical adsorption) and electrical adsorption force (ion exchange adsorption) appear easily between heavy metal ions and inorganic passivators. Heavy metals in feces inhibit microbial activity and hinder anaerobic digestion [24] The content of heavy metals decreases by these forces, which is conducive to the production of biogas and the application of agricultural irrigation [25]. The addition of fly ash also has a positive effect on the on the production of methane [26]. Yang et al. found that methane yield was significantly improved by 25% under the optimal biochar dosage of 5–10% [27]. At present, many scholars have also proved that adding passivator to pig manure could achieve the effective solidification of heavy metals and reduce their pollution to the environment. For example, Kong et al. found that adding calcium magnesium phosphate fertilizer, waste mushrooms and other substrates promoted the humification process of compost and improved the passivation performance of Cu, Zn, Cd, Cr and Pb [28]. Zheng et al. found that the bioavailability of Zn and Cu in pig manure composting process could be reduced by adding a certain proportion of SE [29]. At the same time, scholars also found that modification of passivator [30] and change of particle size [31] could further improve its passivation effect and reduce its bioavailability.

However, at present, many related studies lack the comparison with the single material performance of gas production and the passivation of heavy metals. The research and experimental data in this area are very important for the assessment of heavy metal pollution risk and improvement of gas production. Therefore, this experiment takes fresh pig manure and corn straw as digestive raw materials, and the purpose of this study is as follows: (1) evaluate the effects of different compound passivators on the total amount of biogas produced by the anaerobic digestion of pig manure, the average methane concentration and the passivation effect of Zn and Cu; (2) measure the oxidation-reduction potential and pH of mixed materials and carry out the preliminary study on the changes of physical and chemical properties of mixed anaerobic digestion of pig manure straw; and (3) study the effects of different compound passivators on organic structure changes before and after the anaerobic digestion of pig manure by Fourier transform infrared spectroscopy (FTIR).

## 2. Materials and Methods

### 2.1. Test Raw Materials and Inoculum

In this experiment, fresh pig manure and corn straw were used as raw materials for digestion. Fresh pig manure was collected from a farm in Changzhou. The corn straw was received from a test field in Changzhou, which was crushed after drying and sieved through 200 meshes. The inoculum was the biogas slurry of a large biogas project in Changzhou, which has been running well for more than 3 months. The specific properties of raw materials were presented in Table 1.

### 2.2. Test Design

The self-made small 1L laboratory anaerobic digestion device (Figure 1) was used in the test. The effective volume of the fermentation tank was 800 mL, a water storage bottle and a water collecting bottle were set at the rear and the two bottles were connected with each other through glass tubes and rubber tubes. The liquid in the water storage bottle was squeezed by biogas and discharged into the water collecting bottle when pig manure produced gas in the fermentation tank. To avoid the impact of CO_2_ dissolving in water on gas production, saturated sodium bicarbonate was used rather than tap water. The fermentation tank was placed in a constant temperature water bath to keep the temperature of the fermentation environment constant.

Six groups in the initial test were set. The ratio of fresh pig manure and straw (TS ration) in all groups was different (Table 2). The total dry matter content of the material was 6%, and the residual biogas slurry after inoculation was at least 30%. All tests were carried out in a constant temperature water bath with a digestion temperature of 35 ± 1 °C and a digestion cycle of 30 days. The TS value of the group with the best gas production performance was chosen as the optimal TS ratio.

After determining the optimal TS ratio, 5 groups of pig manure anaerobic composting tests were set up and labeled as test groups L, A, H, Y and CK. The groups of L, A, H and Y indicate that ferric sulfate (FeSO_4_), attapulgite (AT), sepiolite (SE) and ferric oxide (Fe_2_O_3_) with a dry matter mass ratio of 5% were added as passivators for anaerobic digestion, and the CK group was a blank control group without passivators. Three kinds of passivators with better performance (labeled as A, B and C) were introduced into follow-up experiments, and orthogonal tests were arranged according to different proportions (Table 3). Gas production and average methane concentration were taken as the inspection indexes, the optimal addition ratio of the three passivators selected was the inspection factor. When the ratio of pig manure to straw was 8:2, the initial pH value was about 6.4 and the pH in the acid production stage of fermentation dropped significantly. Finally, the pH value gradually rose and finally stabilized at around 7.0.

### 2.3. Index Determination Method

(1)Gas production: the daily gas production of the unit is measured using the drainage method. The methane concentration is measured by using the light interference methane detector (cjg-100), and the methane concentration in biogas is determined by the movement of the light interference derivative line in the instrument.(2)Dry matter mass (TS) and volatile solid content (VS) of pig manure and straw: the pig manure and straw are placed into a constant temperature drying oven for drying at 105 °C and placed in a muffle furnace for high temperature at 600 °C for 4 h. their mass is measured by the differential weight method.(3)Chemical species of heavy metals: the forms of heavy metals in pig manure were extracted by BCR sequential extraction method [32]. After extraction, the contents of various forms of heavy metals were determined by flame atomic absorption spectrometry (AAS). The BCR sequential extraction method was put forward by the European Commission [33]. Heavy metals can be divided into 4 chemical species by BCR sequential extraction method, and the 4 chemical species are defined as follows:

Weak acid extracted state: the state mainly includes the water soluble state, exchangeable state and carbonate combined state, which are easily leached from the target under low pH conditions. As the concentration of water-soluble metal is often lower than the detection limit of the instrument, when calculating the weak acid extracted state, the water-soluble state and exchangeable state are generally combined for calculation.

Reducible state: the state is also called the bound state of iron-manganese oxide, the Fe-Mn bond in it will be destroyed under the conditions of external interference and the heavy metals in this state are easily released into the environment, thus causing harm to the environment.

Oxidizable state: the state mainly includes the organic bound state, which is easily activated under oxidation conditions.

Residual state: heavy metals in the state are mostly found in the mineral lattice of primary ore and secondary ore. Such residual heavy metals are difficult to release into the environment, and their potential biological toxicity is relatively small.

(4)FTIR spectral analysis: the information of its functional groups was obtained by FTIR, which was used to analyze the material before and after anaerobic digestion. Take 1 mg freeze-dried biogas residue and 200 mg anhydrous potassium bromide to mix evenly, and grind thoroughly to fine powder in agate mortar to homogenize the sample. The mixture is then pressed on an oil press at 102 MPa for 5–10 min to form a transparent sheet. Infrared transmission absorption spectra were obtained on NicoletIS50 with a scanning interval of 500–4000 cm^−1^.

### 2.4. Statistical Analysis and Kinetic Equation of Gas Production

The improved Gompertz equation is used to simulate its biogas production process, and the formula is as follows:(1)$$Y=A\times \exp\left\{-\exp\left[\left(B-x\right)/C\right]\right\}$$
where *Y* is the cumulative gas production of biogas, mL/gTS; *A* is the maximum production potential of mixed materials, mL/gTS; *B* is the time when the degradation of mixed materials reaches the peak, d; *x* is the gas production days of mixed materials, d; and *C* is the ratio of the time required for the degradation of the mixed material to e (2.718) when the biogas reaches the maximum yield, d.

The bioavailability of heavy metals before and after anaerobic digestion can be evaluated by the MF value [34], which can be calculated as follows:(2)$$\mathrm{MF}=\frac{c\;\left(F_{1}\right)+\;c\;\left(F_{2}\right)}{c\;\left(F_{1}\right)+\;c\;\left(F_{2}\right)+\;c\;\left(F_{3}\right)+\;c\;\left(F_{4}\right)}\times 100\%$$
where c (F1), c (F2), c (F3) and c (F4) represent the contents of the four heavy metal chemical species. F1–F4 represent acid extraction, reducible, oxidizable and residual states, respectively, in the BCR sequential extraction method.

## 3. Results and Discussion

### 3.1. Gas Production

The gas production of each anaerobic digestion tank was recorded every day and presented in Figure 2a. The overall gas production law of anaerobic digestion rapidly decreased from about 200 mL to about 50 mL in the first 4 days and increased from 5th day. All groups reached the peak value of gas production between the 15th and 18th day. The daily maximum gas production of the P0–P5 group was 342, 467, 555, 567, 544 and 541 mL, respectively. The daily gas production gradually decreased and remained below 100 mL after 18 days.

At the initial stage of digestion, the daily gas production was relatively high, which may be due to the fact that the raw materials used for anaerobic digestion were wrapped in sacks for a short time and the anaerobic digestion itself was carried out at the acid production stage. The accumulation of small molecular acids in pig manure promoted the methane-producing bacteria in the biogas slurry to produce gas. The activity of methanogens was inhibited by the entry of air. Gas production decreased at the initial stage. Complex macromolecular organics in pig manure were decomposed into soluble small molecule organics by the facultative anaerobic microorganisms after 1–2 days. Small molecule organics were further used at the acid production stage [35] and the air in pig manure was consumed during this process. The daily gas production gradually increased until the speed limit period of gas production occurred at about 12 days, which may be due to the accumulation of a large amount of ammonia nitrogen [36]. Microorganisms decomposed the ammonium nitrogen organic matter in pig manure. The high concentration of ammonia inhibited the gas production of methanogens [37].

The time of reaching the maximum daily gas production of P0 and P1 groups was shorter than other groups. The maximum daily gas production of other groups was higher than P0 and P1. Although the addition of straw could lead to the lag of daily peak gas production, the daily gas production could increase. By comparing P2, P3, P4 and P5 groups, it can be seen that the daily gas production needs more time to recover in the acidification stage with the increase in the amount of straw addition. Excessive straw addition might cause the serious acidification of the whole system [38,39]. Inappropriate acidification level exceeds the utilization capacity of methanogens in the acid production stage. The pH of the fermentation system decreased, and methanogens were inhibited for a long time.

The total gas production curve trend (Figure 2b) of different anaerobic digestion tanks was all relatively flat in the early stage, and then increases sharply after 5 days and finally flattens out. Methanogens used short chain organic acids, such as acetic acid, propionic acid, butyric acid and alcohols, to rapidly produced gases in periods of sharp increases of gas production [40]. The total gas production of the 6 test groups is: P3 > P2 > P4 > P1 > P5 > P0. Methanogens could be inhibited by the lower overall pH of the P3 group, and P2 (the ratio of pig manure to straw is 8:2) was selected as the experimental background in the subsequent experiments.

The changes of total gas production and average methanogenic concentration in all groups under the addition of different passivators are shown in Figure 3. The total gas production of the blank (CK) group is 7088 mL, and the average methanogenic concentration is 55.2%. The gas production performance of anaerobic digestion was significantly improved after adding AT and SE as passivators, and the total gas production reached 7821 and 7557 mL, which were higher by 10.34 and 6.62% than the CK group. The average methane production concentration reached 55.9 and 55.5% and was higher by 0.7 and 0.3% than the CK group.

AT and SE can provide good living environments for methanogens [41]. These two substances have a specific crystal structure and rich specific surface area. AT and SE have a good adsorption effect on digestion inhibitory substances (free ammonia, hydrogen sulfide) and can effectively reduce the content of harmful substances in the system [42].

The gas production performance of anaerobic digestion was also improved by the addition of FeSO_4_, while not as effective as Group A and H. The total gas production was 7411 mL, which was 4.56% higher than the CK group. The average methane production concentration was 55.6%. When Fe_2_O_3_ was added as a passivator, the improvement of the gas production performance of anaerobic digestion was the worst. The total gas production was slightly higher than the CK group, while the average methane production concentration even lower than the CK group. Lu et al. found the addition of Fe_2_O_3_ to pig manure for anaerobic digestion enhanced methanogenesis [43], but improper dosages of ferric might reduce the methane production potential [12].

The L9 (33) orthogonal test was carried out based on the passivation effect of previous experiments. The total biogas production and the average methane concentration were taken as the inspection indicators, the addition amount of AT (A), FeSO_4_ (B) and SE (C) were selected as the inspection factors. According to the results of different treatment groups (Table 4), after addition of inorganic compound passivator, all indicators increased after anaerobic digestion. The effect of inorganic passivators on the total gas production was: FeSO_4_ > AT > SE. The additional amount of passivator in the optimal treatment group is 4, 6 and 6 g (Table 5). The effect of inorganic passivators on the total gas production is FeSO_4_ > AT > SE. The amount of passivator added in the optimal treatment group is 6, 2 and 6 g. By comparing the sum of index values of factor B (the amount of FeSO_4_ added) at different levels (Ki, i = 1, 2, 3), high concentration of FeSO_4_ would inhibit gas production. Sulfate-reducing bacteria have higher affinity for acetic acid and hydrogen than methanogens by the accumulation of SO_4_^2−^, and anaerobic digestion was inhibited [44].

Spss22.0 (IBM, New York, NY, USA) was used to analyze the variance of L9 (33) orthogonal test results, there are certain differences in the significance of various indicators under the influence of different factors. The passivation performance of AT, FeSO_4_ and SE were selected as inspection factors. Therefore, according to the experimental results, the optimal ratio of inorganic composite passivator was AT, FeSO_4_ and SE with the ratios of 7.5 g/L, 5 g/L and 7.5 g/L.

### 3.2. Passivation of Cu and Zn

The chemical species distribution of Cu in pig manure changed significantly (Figure 4a) after a cycle (30 days) of anaerobic digestion. Cu in fresh pig manure accounted for a high proportion of weak acid extracted state (32%), reducible state (35%) and oxidizable state (25%). Anaerobic digestion promoted the transfer of Cu in raw materials to the liquid phase and solid phase. The distribution rate of weak acid extraction state and reducible state with poor chemical stability decreased. The reducible state Cu is mainly iron-manganese oxide-bound Cu. Cu forms coordination compounds with the surface of iron-manganese oxide, or Cu is isomorphous in iron-manganese oxide. The extraction state and reducible state changed to an oxidizable state and residue state with strong chemical stability through various chemical precipitation [41]. There are two forms of oxidizable copper, Cu(II) and Cu(I). Cu(II) combines with organic matters, while Cu(I) reacts with S^−1^ to form Cu_2_S [35]. The addition of passivators had a better passivation effect on Cu than the CK group, which indicated that adding iron-based compounds and water-containing magnesium aluminosilicate clay minerals to pig manure strengthened the passivation effect on Cu. With the addition of FeSO_4_, AT, SE and Fe_2_O_3_, the bioavailability of Cu in pig manure in all groups fell to 40.36, 39.85, 44.05 and 48.61%. Among the four passivators, AT (41.87%) had the best passivation effect on Cu in pig manure, which was followed by FeSO_4_ (39.88%) and SE (34.14%), while Fe_2_O_3_ (29.11%) had the worst performance of passivation. The passivation effect of water-containing magnesium aluminosilicate clay minerals on Cu in pig manure was better than iron-based compounds.

Zn in fresh pig manure mainly existed in a weak acid extracted state and reducible state (Figure 4b), and its ratio reached 40 and 45%. After anaerobic digestion, the content of oxidizable state and residual state in pig manure increased, the content of weak acid extraction state decreased and the content of reducible state displayed a slight difference. Zinc has high adsorption affinity for Fe-Mn oxides and hydroxides, so Zn (II) can be adsorbed by these substances to form a reducible state of Zn [45]. Most exchangeable Zn was transformed from an organic combination and carbonate precipitation [46]. With the addition of FeSO_4_, AT, SE and Fe_2_O_3_, the bioavailability of Zn in pig manure in all groups was reduced to 70.61, 67.45, 70.49 and 77.42%. The passivation effect of different inorganic passivators was AT (19.32%) > FeSO_4_ (16.45%) > SE (14.96%) > Fe_2_O_3_ (9.39%). The passivation effect of anaerobic digestion on Zn was not as good as Cu, which may be related to the different affinities of humic acid for Cu and Zn [42]. Liu et al. found that humic acid had a stronger binding ability with Cu through a series of complexation and chelation [6].

The reduction rate of various states of Cu in pig manure after the addition of different inorganic passivators was shown in Figure 5a. Weak acid extracted state heavy metals in pig manure have the worst stability. AT and SE showed an effective passivation performance on weak acid extracted heavy metals, and the reduction rate was 55.63 and 51.76%. Clay minerals have higher specific surface area, developed pore structure and unique crystal structure [1,34]. Forces such as Van der Waals force for physical adsorption, chemical bond force for chemical adsorption and electrical adsorption force for ion exchange adsorption were easy to produce between the adsorbent and adsorbate [47], and the content of weak acid extracted heavy metals was reduced. The addition of FeSO_4_ and Fe_2_O_3_ helped to transform reducible state heavy metals into an oxidizable and residual state, which were more stable, with reduction rates of 38.27 and 22.51%. Attributed to the high specific surface area and strong redox capacity of iron-based compounds, more stable amorphous precipitation and insoluble secondary minerals were formed by the adsorption between iron-based compounds and heavy metals [48]. Ferric in iron-based compounds could also be used as the electron donors of heavy metal ions, and insoluble heavy metal compounds were formed through oxidation-reduction [49].

The reduction rate of various states of Zn in pig manure after the addition of different inorganic passivators is shown in Figure 5b. Because of the special crystal structure of water-containing magnesium aluminosilicate clay minerals, AT and SE still showed a high passivation effect on weak acid extracted Zn, with a reduction of 45.21% and 37.11%. The content of the reducible state increased after anaerobic digestion, which is similar to the passivation effect of moderate temperature anaerobic digestion of pig manure and cow manure on reducible Zn studied by Jin et al. [50]. Among all chemical species of heavy metals, the increase ratio of the oxidizable state was the highest, reaching 114.12 (FeSO_4_), 113.49 (AT), 96.46 (SE), 67.41 (Fe_2_O_3_) and 50.87% (blank). Microorganisms decomposed macromolecular organics to humus through a series of biochemical reactions, a large amount of humus could combine with oxidizable heavy metals. The content of the oxidizable state in heavy metals increased. The most stable residual heavy metals have a certain increase compared with the blank group. More residual metals are solidified in the mineral lattice and are difficult to released into the natural environment.

The L9 (33) orthogonal test was carried out based on the passivation effect of previous experiments. The passivation effect on Zn and Cu were taken as the inspection indicators, the additional amounts of AT (A), FeSO_4_ (B) and the SE (C) were selected as the inspection factors. The anaerobic digestion test of pig manure was carried out according to the design scheme in Table 6. The effect of inorganic passivators on the passivation effect of Zn and Cu is AT > FeSO_4_ > SE. The additional amount of passivator in the optimal treatment group is 6, 4 and 6 g, which, expressed in ratio, is 7.5 g/L, 5 g/L and 7.5 g/L. By comparing the sum of index values of factor B (the amount of FeSO_4_ added) at different levels (Ki, i = 1, 2, 3), the high concentration of FeSO_4_ inhibits the performance of passivation. The excessive iron content in FeSO_4_ might lead to the increase of biological cytotoxicity [51]. Spss22.0 was used to analyze the variance of L9 (33) orthogonal test results. Table 7 shows that there are certain differences in the significance of various indicators under the influence of different factors. The optimal ratio of inorganic composite passivator was AT, FeSO_4_ and SE with a ratio of 7.5 g/L, 5 g/L and 7.5 g/L.

### 3.3. Fourier Infrared Spectrogram Analysis

The change of material functional groups before and after anaerobic digestion was analyzed by FTIR (Figure 6). In terms of the relative intensity of absorption peaks, there were slight differences in the spectral characteristics of pig manure before and after anaerobic digestion in all treatment groups. The addition of inorganic composite passivator changed organic matter in pig manure and straw during the anaerobic digestion period [52,53]. The vibration peaks at 3408~3440 cm^−1^ can be regarded as -OH stretching vibration peaks in carbohydrates. The vibration peaks at 2850~2930 cm^−1^ were assigned as C-H stretching vibration peaks, which might belong to methylene of aliphatic compounds. This peak intensity decreased after anaerobic digestion, the content of these substances in pig manure decreased after anaerobic digestion [53,54,55]. Because microorganisms cannot directly metabolize carbohydrates, proteins, fats and other biological macromolecules. These substances are hydrolyzed into small molecular organics by the action of hydrolytic enzymes in microorganisms. Small molecular organics can be used by acidogenic bacteria and hydrogen-producing bacteria to provide food and a suitable living environment for methanogens [56]. The vibration peaks at 1620~1650 cm^−1^ can be listed as the C=C stretching of aromatics and olefins and can also be considered as -COO- stretching and C=O stretching vibration peaks of carboxylic acid compounds. The vibration peak at 1010~1050 cm^−1^ is considered as the aromatic group absorption peak. The increase of the relative intensity of these absorption peaks indicated that organic materials, such as cellulose and lignin, were transformed to olefins and aromatic ring humus after anaerobic digestion. The content of humus in biogas residue increased after adding inorganic composite passivators. Humus had a negative charge [57], a large number of carboxyl and alcohol hydroxyl groups. Humus could combine with heavy metal ions, which was conducive to the passivation of heavy metals. To sum up, adding different amounts of inorganic compound passivator to the system of anaerobic digestion can effectively promote the metabolic activities of acidogenic bacteria and methanogens and reduce the content of macromolecular organics.

## 4. Conclusions

The gas production efficiency of anaerobic digestion with addition of corn straw and passivator was better than single materials. The problem of insufficient gas production can be dealt with through the promotion of microbial activity using mixed materials. Heavy metals in pig manure can also be better passivated during anaerobic digestion with the addition of mixed material. When the ratio of pig manure to straw is 8:2, adding an inorganic passivator could effectively increase gas production and the passivation of Cu and Zn performance during anaerobic digestion of pig manure. The passivation performance of the composite passivator for Cu and Zn was better than a single passivator, the optimal ratio of passivators for AT, FeSO_4_ and SE is 7.5 g/L, 5 g/L and 7.5 g/L. The vibration peak intensity of hydroxyl and carbon-hydrogen bonds in the pig manure residue decreased after adding passivation, which were shown in the FTIR spectrum, the decrease indicated that macromolecular organic matter was decomposed. The vibration peak intensity of carboxylic acid and carbonyl increased. The content of humus increased, and heavy metals in pig manure were easily combined with humus. The phenomenon improved the passivation efficiency of heavy metals in the anaerobic digestion process.

Energy stress was relieved by the improvement of gas production performance, and biogas residue was used as safer fertilizer in farmland irrigation by the efficient passivation of Cu and Zn. This study can provide a theoretical basis for the safe application of biogas fertilizers. This technology deserves the large-scale promotion with the above advantages and cheap material sources. However, the actual environment is more complex than the laboratory, which leads to uncertainty regarding the actual benefits of composite passivators on the anerobic digestion. More experiments on the application in industry and agriculture need to be carried out to prove the actual impact of composite passivators on the environment. In addition, for industrial production, the preparation conditions of composite passivator need to be further optimized according to different environmental conditions and requirements.

## Figures and Tables

**Figure 1 ijerph-19-14094-f001:**
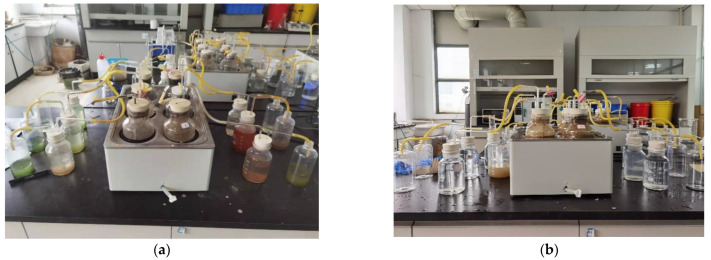
The top (**a**) and front (**b**) of self-made laboratory anaerobic digestion device.

**Figure 2 ijerph-19-14094-f002:**
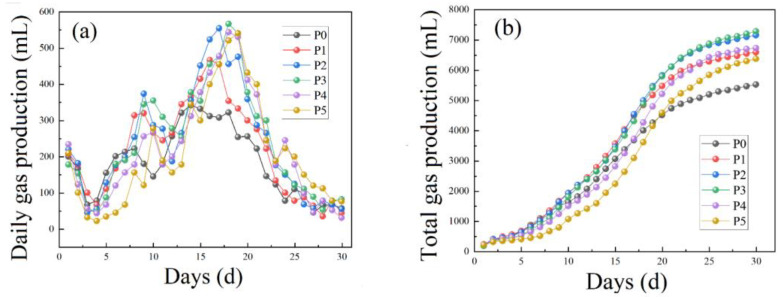
(**a**) Daily gas production and (**b**) total gas production of THE mixed anaerobic digestion of pig manure and straw in different proportions.

**Figure 3 ijerph-19-14094-f003:**
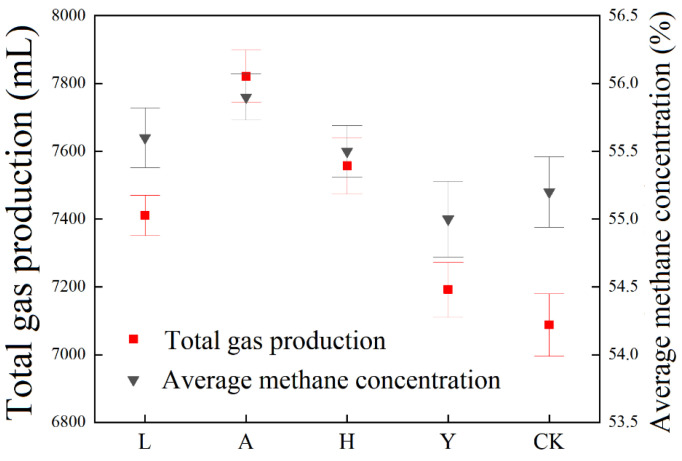
Effect of different inorganic passivators on gas production performance.

**Figure 4 ijerph-19-14094-f004:**
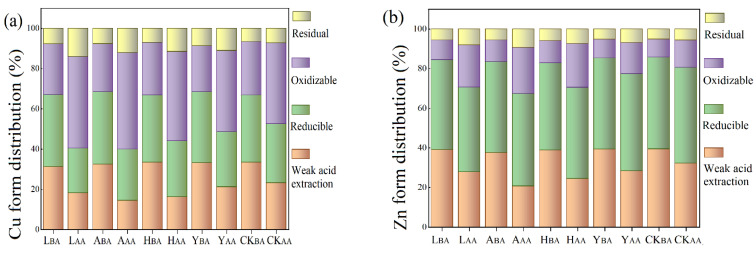
Effects of different inorganic passivators on the distribution of the form of (**a**) Cu and (**b**) Zn. BA: before anaerobic digestion; AA: after anaerobic digestion.

**Figure 5 ijerph-19-14094-f005:**
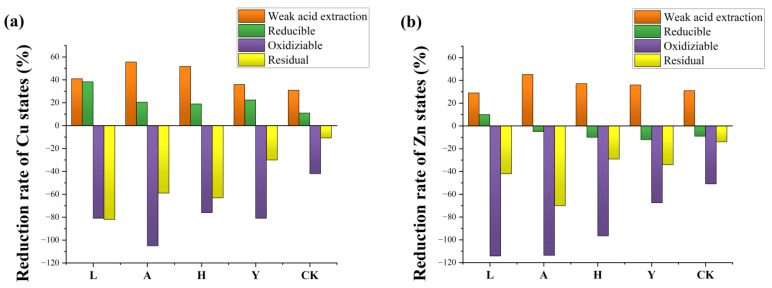
Reduction rates of various forms of (**a**) Cu and (**b**) Zn after different inorganic passivation treatments.

**Figure 6 ijerph-19-14094-f006:**
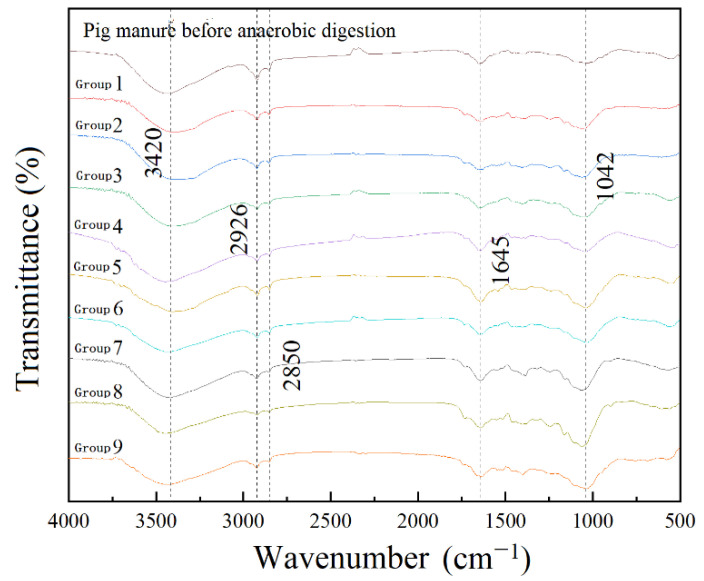
FTIR spectra of pig manure before and after anaerobic digestion with different compound passivators.

**Table 1 ijerph-19-14094-t001:** Nature of raw materials.

Test Materials	MC * (%)	DF * (%)	VF * (%)	MF * (%)	NF * (%)	C/N *
fresh pig manure *	79.84	20.16	13.59	9.2	0.6	15.33
raw straw	8.75	91.25	81.12	36.41	0.55	66.20

* MC: the moisture content; DF: dry matter mass fraction; VF: volatile solid mass fraction; MF: mass fraction of organic carbon; NF: nitrogen mass fraction; C/N: carbon nitrogen ratio.

**Table 2 ijerph-19-14094-t002:** Experimental designs.

Test Group	P/S *	C/N
P0	10:0	15.33
P1	9:1	20.04
P2	8:2	24.82
P3	7:3	29.78
P4	6:4	34.63
P5	5:5	39.66

* P/S: pig manure to straw ratio.

**Table 3 ijerph-19-14094-t003:** L9 (33) factor level table of orthogonal test.

Level	A, g	B, g	C, g
1	2	2	2
2	4	4	4
3	6	6	6

**Table 4 ijerph-19-14094-t004:** Total biogas production and average methane concentration in biogas residue from pig manure anaerobic digestion.

Test Number	A	B	C	TBP */mL	AMC */%
1	1	1	1	7038.32	56.21
2	1	2	2	7258.97	56.83
3	1	3	3	6534.41	55.52
4	2	1	2	7552.48	57.69
5	2	2	3	7767.44	58.42
6	2	3	1	6131.37	56.47
7	3	1	3	8216.61	60.77
8	3	2	1	7826.35	58.91
9	3	3	2	7273.53	57.36

* TBP: total biogas production; AMC: average methane concentration.

**Table 5 ijerph-19-14094-t005:** Range analysis of total biogas production and average methane concentration in biogas residue from pig manure anaerobic digestion.

Project		A	B	C
TBP	K1	6943.90	7602.47	6998.68
	K2	7150.43	7617.59	7361.66
	K3	7772.16	6646.44	7506.15
	Range R	828.26	971.15	507.47
	PSF *		B, A, C	
	The optimal solution		A3 B2 C3	
AMC	K1	56.19	58.22	57.20
	K2	57.53	58.05	57.29
	K3	59.01	56.45	58.24
	Range R	2.83	1.77	1.04
	PSF		A, B, C	
	The optimal solution		A3 B1 C3	

* PSF: Primary and secondary factors.

**Table 6 ijerph-19-14094-t006:** Passivation effect of Zn and Cu in biogas residues from pig manure anaerobic digestion.

Test Number	A	B	C	ZnPE */%	CuPE */%
1	1	1	1	18.54	42.11
2	1	2	2	25.72	44.84
3	1	3	3	23.68	45.25
4	2	1	2	24.37	50.52
5	2	2	3	27.91	58.97
6	2	3	1	23.56	47.84
7	3	1	3	28.49	54.53
8	3	2	1	28.25	58.57
9	3	3	2	26.83	53.96

* ZnPE: the effect of Zn passivation; CuPE: the effect of Cu passivation.

**Table 7 ijerph-19-14094-t007:** Range analysis of the passivation effect of Zn and Cu in the biogas residue from pig manure anaerobic digestion.

Project		A	B	C
ZnPE	K1	22.65	23.80	23.45
	K2	25.28	27.29	25.64
	K3	27.86	24.69	26.69
	Range R	5.21	3.49	3.24
	PSF		A, B, C	
	The optimal solution		A3 B2 C3	
CuPE	K1	44.07	49.05	49.51
	K2	52.44	54.13	49.77
	K3	55.69	49.02	52.92
	Range R	11.62	5.11	3.41
	PSF		A, B, C	
	The optimal solution		A3 B2 C3	

## Data Availability

Not applicable.

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
