# Peer review of "Effects of Inorganic Passivators on Gas Production and Heavy Metal Passivation Performance during Anaerobic Digestion of Pig Manure and Corn Straw"

_ijerph, 2022, doi:10.3390/ijerph192114094_

Round 1

Reviewer 1 Report

The topic is useful, and the results given in the paper have practical importance. Basically,  the experiments are planned well, and the explanations are mainly correct. Some points, however, should be improved, and some corrections in the text should also be made.

For example: 

1., The name sepiolite and attapulgite should be given in the abstract and the first place in the text to solve the abbreviations SE And AT. 

2., Abbreviation "TS" (line 137) should be solved. 

3., Ferric sulfate is non-capital F (line 139)

4., Chemical species should be defined (e.g., extractable with... and .. with ..) (line 164)

5., 102 Pa is not an overpressure; would it be 10E2 MPa? (line 170)

6., where is not capital W (line 178)

7., Please define what mean "reducible" and "oxidizable" states of Cu .(Cu(II) and Cu(I)? ) (line 279-281)

8.,  Zn might be extractable with weak acids; the "weak-acid extracted state" is unusual because the chemical state of Zn is Zn(II) in either weak or strong acid extracted form (independently that salt-like or complexed Zn(II)). (line 294)

9., What is the reducible state of Zn? Zn(II)? It is the one stable form of Zn and, really, can be reduced, but only into metallic Zn, which is not a probable form in an aq. environment. (line 294). 

10., The IR evaluation is not completely correct. In a multi-component system, as manure, compounds are able to make hydrogen bonds, it can be said only, that the bands around 3408-3440 might belong mainly to OH groups (and not amide), and the NHs are expected around 3100-3200 cm-1. The CH modes may belong to CH3 groups as well, not only CH2 (lines 360-370). 

These assignments should be given in a more general form because the shift of each type of band can give coinciding with other types. In principle, the wide bands might be decomposed with curve anaysis, but due to the presence of many components, it has no meaning.  

11., Carboxylic should not be given with capital (C) (line 376)

12., The pH (Acidity) should be given for the starting/fermented mixtures because the behavior of Fe2O3 and FeSO4 mainly depends on pH. FeSO4 hydrolyzes in neutral/alkaline pH with the formation of Fe(OH)2, and that oxidizes in the presence of traces of O into FeOOH. Depending on pH, Fe(II)+ Cu(II)= Cu(I) + Fe(III) (equilibrium) should be appeared. In order to disclose/confirm the presence of these processes, the pH should be taken into consideration. 

Similarly, the ion exchange capicty towards Cu and Zn and the selectivity of SE and AT also depend on pH. 

Author Response

Response to Reviewer 1 Comments

Point 1: The name sepiolite and attapulgite should be given in the abstract and the first place in the text to solve the abbreviations SE And AT.

Response 1: Thank you for your suggestion. The full names were given when the abbreviations first appeared according to your suggestion. We have modified across whole manuscript to supplement full names for all abbreviations according to your suggestion. (Line 12-13, P1, in red)

Point 2: Abbreviation "TS" (line 137) should be solved.

Response 2: Thank you for your suggestion. Yes, the meaning of TS did not clearly explain in this article. We had explained the meaning of TS and the way of determining the best TS value according to your suggestion. (Line 147-152, P4, in red)

Point 3: Ferric sulfate is non-capital F (line 139).

Response 3: Thank you for your suggestion. Yes, Ferric sulfate is non-capital. We had capitalized the first letter of this word according to your suggestion. (Line 156, P4, in red)

Point 4: Chemical species should be defined (e.g., extractable with... and .. with ..) (line 164).

Response 4: Thank you for your suggestion, a definition of the chemical species of heavy metals had been added after the section in the article. The harm degree of different species of heavy metals was also briefly explained according to your suggestion. (Line 183-200, P5,in red)

Point 5: 102 Pa is not an overpressure; would it be 10E2 MPa? (Line 170)

Response 5: Thank you for your suggestion. After verification, the output pressure of the hydraulic press in the FTIR sample making process is 102Mpa, which has been corrected in the manuscript according to your suggestion. (Line 205, P5)

Point 6: Where is not capital W (line 178).

Response 6: Thank you for your suggestion. We had changed w to lowercase according to your suggestion according to your suggestion. (Line 211&218, P6)

Point 7: Please define what mean "reducible" and "oxidizable" states of Cu. (Cu(II) and Cu(I)?) (Line 279-281)

Response 7: Thank you for your suggestion, "reducible" and "oxidizable" states of Cu have been explained in this paper. There are two forms of oxidizable copper, Cu(â…¡) and Cu(â… ). Cu(â…¡) will combine with organic matters, while Cu(â… ) will react with S-1 to form Cu2S according to your suggestion.. (Line 318-324, P9)

Point 8: Zn might be extractable with weak acids; the "weak-acid extracted state" is unusual because the chemical state of Zn is Zn(II) in either weak or strong acid extracted form (independently that salt-like or complexed Zn(II)). (line 294)

Response 8: Thank you for your suggestion. As the concentration of water-soluble metal is often lower than the detection limit of the instrument, the value of weak-acid extracted state is generally calculated by combining water-soluble and exchangeable states. Therefore, most of the extracted weak acids may be water-soluble. The explanation has been supplemented in the previous part of the article when introducing the four states of heavy metals according to your suggestion. (Line 186-191, P5, in red)

Point 9: What is the reducible state of Zn? Zn(II)? It is the one stable form of Zn and, really, can be reduced, but only into metallic Zn, which is not a probable form in an aq. environment.

Response 9: Thank you for your suggestion. The reducible state in BCR continuous extraction method refers to the combined state formed by the reaction and complexation of heavy metals in water with hydrated iron oxide and manganese oxide, not just the reduced zinc. Zinc has high adsorption affinity for iron manganese oxides and hydroxides, so Zn (II) will be adsorbed by these substances to form reducible state of Zn. The explanation has been added in manuscript according to your suggestion. (Line 336-339, P9, in red)

Point 10: The IR evaluation is not completely correct. In a multi-component system, as manure, compounds are able to make hydrogen bonds, it can be said only, that the bands around 3408-3440 might belong mainly to OH groups (and not amide), and the NHs are expected around 3100-3200cm-1. The CH modes may belong to CH3 groups as well, not only CH2 (lines 360-370). 

These assignments should be given in a more general form because the shift of each type of band can give coinciding with other types. In principle, the wide bands might be decomposed with curve anaysis, but due to the presence of many components, it has no meaning.

Response 10: Thank you for your suggestion. The interpretation of -OH and -NH groups had been changed in the article. The attribution of other groups was also listed the possibility rather than the certainty to avoid disputes. These assignments were given in a more general form according to your suggestion. (Line 407-416, P12, in red)

Point 11: Carboxylic should not be given with capital (C) (line 376)

Response 11: Thank you for your suggestion. The c of Carboxylic had been changed to lowercase according to your suggestion. (Line 419, P12, in red)

Point 12: The pH (Acidity) should be given for the starting/fermented mixtures because the behavior of Fe2O3 and FeSO4 mainly depends on pH. FeSO4 hydrolyzes in neutral/alkaline pH with the formation of Fe(OH)2, and that oxidizes in the presence of traces of O into FeOOH. Depending on pH, Fe(II)+ Cu(II)= Cu(I) + Fe(III) (equilibrium) should be appeared. In order to disclose/confirm the presence of these processes, the pH should be taken into consideration.

Similarly, the ion exchange capicty towards Cu and Zn and the selectivity of SE and AT also depend on pH.

Response 12: Thank you for your suggestion. In the previous experiment, PH was at a medium low level in the whole anaerobic digestion process. The pH value of this background has been added in the manuscript according to your suggestion. (Line 162-165, P4, in red)

Reviewer 2 Report

Title

            The title of the manuscript is ok but in my opinion the author should use the word corn straw instead of straw because the author done his work by the utilization of this straw, after this addition, it looks specified and attractive, while the other stuff is ok.

Graphical abstract & abstract

                                                            The abstract is ok but for the enhancement of its citations and downloads, the author must add its graphical abstract which attracts the attentions of the readers.                        

Introduction

                        In the part of introduction, before the reference number 15 the author wrote the word ultrasonic instead of writing ultra-sonication. After the reference number 17, the author was explaining about certain forces like [Van der Waals force (physical adsorption), chemical bond force (chemical adsorption) and electrical adsorption force (ion exchange adsorption) are easy appear between adsorbent and adsorbent], the author before submitting the final manuscript must explain what does adsorbent and adsorbent indicates.

The author explained all the stuff very precisely, but he didn’t explain the adverse effects of Heavy metals on living organisms. This thing must be added by the side of the author by utilizing the below doi links which could be proved very beneficial for this manuscript.

i)                   10.1186/s40816-018-0093-8

ii)                 10.1016/j.biortech.2007.01.057

iii)               10.1007/3-540-45838-7_1    etc.

If the author already utilized the above articles in the manuscript than he/she needs to add the above and below mentioned stuff with the assistance of other sources or research articles etc.

Materials and methods

                                    Information written about the materials is quite satisfactory. All the section are f9 in t section 2.4, the author used wrong spells of analyses instead of analysis. In the last sub-section, the author was talking about 4 heavy metals and forgot the addition of their names, it needs to be modified by the author of the manuscript.

Results and Discussion

                                                The author of the manuscript explains the results in a very systematic way. All the information given by the author is satisfactory but it will look more attractive if the author add couple of high definition images of his experimental work in the manuscript.

Conclusion

                   Need to add future prospects in last lines while the other part is ok.

Views of the reviewer

                                          All the stuff written by the author is satisfactory but at some of the places that are mentioned above some minor errors are available like not availability of the graphical abstract, images of the experiments, pros and cons of heavy metals in the part of introduction, spells error in the heading and incomplete information about biochar etc. after these minor amendments and the addition of more information from the above mentioned doi’s, the manuscript will be refined and publishable in this journal.

Author Response

Response to Reviewer 2 Comments

Point 1: The title of the manuscript is ok but in my opinion the author should use the word corn straw instead of straw because the author done his work by the utilization of this straw, after this addition, it looks specified and attractive, while the other stuff is ok.

Response 1: Thank you for your suggestion. The straw in the title of the article has been changed to corn straw according to your suggestion. (Line 4, P1, in red)

Point 2: The abstract is ok but for the enhancement of its citations and downloads, the author must add its graphical abstract which attracts the attentions of the readers.

Response 2: Thank you for your suggestion. A graphical abstract has been added to the manuscript according to your suggestion. (Line 27-28, P1, in red)

Point 3: In the part of introduction, before the reference number 15 the author wrote the word ultrasonic instead of writing ultra-sonication. After the reference number 17, the author was explaining about certain forces like [Van der Waals force (physical adsorption), chemical bond force (chemical adsorption) and electrical adsorption force (ion exchange adsorption) are easy appear between adsorbent and adsorbent], the author before submitting the final manuscript must explain what does adsorbent and adsorbent indicates.

Response 3: Thank you for your suggestion. the ultrasonic before [15] has been replaced by ultra-sonication, and the objects of various forces after [17] have also been described according to your suggestion. (Line 87-95, P3, in red)

Point 4: The author explained all the stuff very precisely, but he didn’t explain the adverse effects of Heavy metals on living organisms. This thing must be added by the side of the author by utilizing the below doi links which could be proved very beneficial for this manuscript.

  1. i) 10.1186/s40816-018-0093-8
  2. ii) 10.1016/j.biortech.2007.01.057

iii) 10.1007/3-540-45838-7_1 etc.

If the author already utilized the above articles in the manuscript than he/she needs to add the above and below mentioned stuff with the assistance of other sources or research articles etc.

Response 4: Thank you for your suggestion. The toxicity of heavy metals (especially copper and zinc) to organisms has been supplemented in the article (Line 44-53), and the relevant contents in the literature you recommended have been referred to according to your suggestion ([9], [13], P2, [24], P3, in red).

Point 5: Information written about the materials is quite satisfactory. All the section are f9 in t section 2.4, the author used wrong spells of analyses instead of analysis. In the last sub-section, the author was talking about 4 heavy metals and forgot the addition of their names, it needs to be modified by the author of the manuscript.

Response 5: Thank you for your suggestion. The misspelling of analysis in the title of section 2.4 has been corrected. (Line 208, P6, in red) The heavy metal forms represented by F1 to F4 have been supplemented in the manuscript according to your suggestion. (Line 219-220, P6, in red)

Point 6: The author of the manuscript explains the results in a very systematic way. All the information given by the author is satisfactory but it will look more attractive if the author add couple of high definition images of his experimental work in the manuscript.

Response 6: Thank you for your suggestion. Two images of experimental work have been added to the manuscript according to your suggestion. (Line 146, P4, in red)

Point 7: Need to add future prospects in last lines while the other part is ok.

Response 7: Thank you for your suggestion. The future prospect of this technology has been added in the last part of the manuscript according to your suggestion. (Line 450-460, P13, in red)

Point 8: All the stuff written by the author is satisfactory but at some of the places that are mentioned above some minor errors are available like not availability of the graphical abstract, images of the experiments, pros and cons of heavy metals in the part of introduction, spells error in the heading and incomplete information about biochar etc. after these minor amendments and the addition of more information from the above mentioned doi’s, the manuscript will be refined and publishable in this journal.

Response 8: Thank you for your suggestion. Graphic abstract, experimental pictures, heavy metal hazards to biology and other contents have been added to the text, spelling errors have been corrected. Biochar is not included in the four inorganic additives mainly explored in this manuscript, so biochar is not introduced in detail. Relevant contents of the above mentioned Doi have been added to the manuscript according to your suggestion.

Round 2

Reviewer 1 Report

All the reviewer's suggestions have been accepted and the manuscript has been revised, thus I can suggest accepting it to publish as it is.